# Dysbiosis Signature of Fecal Microbiota in Patients with Pancreatic Adenocarcinoma and Pancreatic Intraductal Papillary Mucinous Neoplasms

**DOI:** 10.3390/biomedicines12051040

**Published:** 2024-05-08

**Authors:** Theodoros Sidiropoulos, Nikolas Dovrolis, Hector Katifelis, Nikolaos V. Michalopoulos, Panagiotis Kokoropoulos, Nikolaos Arkadopoulos, Maria Gazouli

**Affiliations:** 14th Department of Surgery, Attikon University Hospital, National and Kapodistrian University of Athens, 12462 Athens, Greece; theosidiropoulos@hotmail.com (T.S.); nmichal@med.uoa.gr (N.V.M.); kokoropoulos@yahoo.gr (P.K.); narkadopoulos@med.uoa.gr (N.A.); 2Laboratory of Biology, Department of Basic Medical Sciences, Medical School, National and Kapodistrian University of Athens, 11527 Athens, Greece; ndovroli@med.uoa.gr (N.D.); katifel@med.uoa.gr (H.K.)

**Keywords:** cancer, IPMN, microbiome pancreas, PDAC

## Abstract

Pancreatic cancer (PC) ranks as the seventh leading cause of cancer-related deaths, with approximately 500,000 new cases reported in 2020. Existing strategies for early PC detection primarily target individuals at high risk of developing the disease. Nevertheless, there is a pressing need to identify innovative clinical approaches and personalized treatments for effective PC management. This study aimed to explore the dysbiosis signature of the fecal microbiota in PC and potential distinctions between its Intraductal papillary mucinous neoplasm (IPMN) and pancreatic ductal adenocarcinoma (PDAC) phenotypes, which could carry diagnostic significance. The study enrolled 33 participants, including 22 diagnosed with PDAC, 11 with IPMN, and 24 healthy controls. Fecal samples were collected and subjected to microbial diversity analysis across various taxonomic levels. The findings revealed elevated abundances of Firmicutes and Proteobacteria in PC patients, whereas healthy controls exhibited higher proportions of Bacteroidota. Both LEfSe and Random Forest analyses indicated the microbiome’s potential to effectively distinguish between PC and healthy control samples but fell short of differentiating between IPMN and PDAC samples. These results contribute to the current understanding of this challenging cancer type and highlight the applications of microbiome research. In essence, the study provides clear evidence of the gut microbiome’s capability to serve as a biomarker for PC detection, emphasizing the steps required for further differentiation among its diverse phenotypes.

## 1. Introduction

Pancreatic ductal adenocarcinoma (PDAC) represents the seventh leading cause of death due to cancer, and in 2020 there were approximately 500,000 new cases [1]. At the same time, patient outcomes among the different types of cancer are consistently low [2], while the survival rates between 1975 and 2011 have only risen from 0.9% to 4.2% for all stages of the disease [3].

Several reasons have been recognized for these meager survival rates. A key factor is delayed diagnosis (in the vast majority of cases, symptoms are either absent or non-specific at early stages), which is typically achieved when metastatic disease is already present in one out of two patients [4]. Moreover, the therapeutic approaches are also accompanied by increased morbidity and mortality. Only 20% of patients with pancreatic cancer are eligible for surgery [5], and pancreaticoduodenectomy, the treatment of choice for PDAC, in the head of the pancreas has a perioperative morbidity that surpasses 50% [6] and a median survival of 25 months [7]. Another important parameter is the complex biology of the PDAC microenvironment, which is desmoplastic and characterized by hypovascularity, resulting in hypoxia and thus poor drug delivery [8]. These characteristics explain the lack of efficiency for both conventional chemotherapeutic approaches and radiation treatments, which tend to offer marginal benefits [9]. 

To counter these issues, extensive research in PDAC genetics has been performed for the identification of diagnostic biomarkers [10,11]. Although a better understanding of the disease’s pathophysiology has been achieved [12,13], the contribution of genetics in early diagnosis or more effective treatments has not been translated into a substantial change regarding its management [14]. 

Current approaches for PDAC early diagnosis have focused on the identification of patients with a high risk of developing the disease. These include lesions that serve as precursors of PDAC, as in the case of intraductal papillary mucinous neoplasms (IPMNs). Identifying these high-risk lesions offers an invaluable opportunity to treat the patient who bears these direct PDAC precursors. Even though IPMNs tend to be asymptomatic, during random abdominal imaging, cysts (the majority of which are IPMNs) are revealed in almost 1 out of 10 patients [15]. However, not all IPMN cases have an equal chance of progressing to PDAC; different types of IPMNs have different progression rates to PDAC, which, in some cases, can be as high as 90% [16]. For that reason, several types of microbiota (including pancreatic [17], salivary [18], and fecal [19,20,21]) are currently being investigated for a possible microbiome signature that would allow the recognition of patients with a high risk of developing PDAC. 

The fecal microbiome offers the advantage of being easily accessible without the need for any invasive procedure, while it is currently widely accepted that it is involved in tumorigenesis of various cancers, including pancreatic [22,23] and distant bile ductal [24], which share developmental origins. Microbiome signatures have already shown promising results in the early diagnosis of cancers, including hepatocellular carcinoma [25] and colorectal cancer [26,27] and simpler pancreatic condition like pancreatitis [28]. More importantly, the fecal microbiome has been already found to play a role in the development of malignancies from precursor lesions, as in the case of colorectal cancer development from polyps [29].

However, several parameters need to be considered during the study of the microbiome. Fecal microbiota are known to be affected by ethnic differences [30]. Several factors, including nutrition and geography, affect the microbiota [31] and provide the rationale for study designs that focus on ethnicity. Moreover, PDAC patients also have other comorbidities and risk factors like alcohol consumption, antibiotics use, blood group, BMI, and oral health, which can affect both PDAC progression [32,33,34,35,36] and the gut microbiota composition [37,38,39,40,41] and need to be addressed in fecal microbiome research.

In the present study, we aim to investigate the dysbiosis signature of the fecal microbiome in PC patients and possible differentiations between PDAC and IPMN. We deem that these changes could serve as tools in the early detection of the disease, contributing to the amelioration of PC prognosis.

## 2. Materials and Methods

### 2.1. Participants

Thirty-three patients, twenty-two patients diagnosed with PDAC, eleven patients diagnosed with IPMN, and twenty-four healthy controls, were enrolled in this study. Patients and controls were prospectively recruited between June 2022 and July 2023 from the “Attikon” General University Hospital, Athens, Greece. Subjects who were newly diagnosed with PDAC or IPMN were recruited prior to any cancer treatment. Patients’ PDAC and IPMN diagnosis was based on a histology report, which was preoperative—for 1 patient with IPMN on active surveillance and for 4 patients with either locally advanced or metastatic PDAC that was inoperable—or postoperative from the surgical resected tissue, which was the case for 3 patients with intraoperative biopsy that were not able to be operated on and 25 that underwent pancreatectomy. Additionally, in all cases, the diagnosis was confirmed with computed tomography (CT) scan, and, for most, magnetic resonance imaging (MRI) and/or magnetic resonance cholangiopancreatography (MRCP) was also performed. For unresectable patients, pathological examinations by endoscopic ultrasound-guided fine needle aspiration (EUS-FNA) were conducted. Controls were matched for age, gender, and hospital where inpatients admitted for diagnoses unrelated to PDACor IPMN were selected from. There were no statistically significant differences between patient and control’s mean age and gender ratio (*p* ≥ 0.05). No dietary restrictions were imposed prior to this study. The inclusion criteria were as follows: age >18 years old, primary-care treatment-naïve patients. Exclusion criteria for all subjects included irritable bowel disease, celiac disease, other cancers, pancreatitis, and autoimmune diseases, as well as any usage of antibiotics, antifungals, probiotics, or prebiotics for at least 3 months prior to sampling. The clinicopathological characteristics are presented in Table 1.

The study followed the ethical principles of the World Medical Association Declaration of Helsinki and was approved by the Institutional Review Board of “Attikon” General University Hospital (644/25-11-2021). All participants provided written informed consent.

### 2.2. Fecal Sample Collection and DNA Extraction

Fecal samples from patients and controls were obtained by Fecal Swab Collection and Preservation System (Norgen BioTek Corp., Thorold, ON, Canada) and stored in the preservative provided at −20 °C until DNA extraction according to the manufacturer’s instructions. Fecal microbial DNA was purified from the fecal samples using the Stool DNA Isolation Kit (Norgen BioTek Corp., Thorold, ON, Canada) following the kit’s instructions. 

### 2.3. Sequencing and Read Processing

Sequencing on the samples was carried out by Eurofins Genomics Europe Sequencing GmbH (Jakob-Stadler-Platz 7, 78467, Constance, Germany) on an Illumina MiSeq platform producing paired-read samples of 300 bp read length based on the V3-V4 amplicons (primers 515F [Parada] FWD:GTGYCAGCMGCCGCGGTAA—806R [Appril] REV:GGACTACNVGGGTWTCTAAT). Raw sequences (average: 185,000 reads per sample of which 80% were high quality with Q > 30 [average length 283nt]) were quality controlled using CUTADAPT v2.7 [42], barcodes were removed using fastp v0.20.0 [43], and reads were merged (97% merging rate) with FLASH v. 2.2.00 [44]. Quality-controlled sequences were used as input to QIIME2 v.2023.5 [45], on which they were denoised and clustered into ASVs (Amplicon Sequence Variants) using DADA2 [46]. Taxonomic classification of ASVs was conducted using the SILVA database v.139 [47] on 99% similarity. All samples were included in these steps. Due to multiple sequencing runs, the final ASV table was processed for batch correction using the “combat” method of the sva R package [48]. All relevant resulting abundance, metadata, and taxonomic assignment files are provided in Appendix A. 

### 2.4. Downstream Bioinformatics Analysis

The downstream analysis of the study was twofold. The first analysis grouped the samples into two categories: pancreatic cancer (PC) samples and healthy control (HC) samples. The second analysis used only the PC samples categorized according to their tumor type into IPMN and PDAC samples. All data underwent normalization steps before subsequent analyses, keeping ASVs with at least 4 counts per sample and a prevalence in at least 10% of the samples. In addition, all ASV counts were rarified to the minimum library size of 9358 for the analyses that could benefit from it and were scaled using Total Sum Scaling (TSS), a method to convert a set of numbers into a single score by adding up all the values and then dividing by the total number of values, which helps to simplify complex data and make it easier to compare and analyze.

Alpha-diversity Shannon and Chao1 indices were applied to the raw bacterial counts to calculate species evenness and observed richness, indicating how many bacterial communities can be detected and how evenly distributed those populations are. Chao1 was based on richness, where the Shannon index accounted for both. Statistical difference between groups was calculated with the Mann–Whitney non-parametric test. False discovery rate (FDR) adjusted *p*-values were calculated with the Benjamini–Hochberg method. Beta-diversity provides a measure of how different the composition of the microbiome is in each sample and group, compared to the rest. Dissimilarities between groups were analyzed and visualized using non-metric multidimensional scaling (NMDS) [49], whereas their statistical power was calculated using Analysis of Similarities (ANOSIM) [50]. 

To identify statistically significant differences in the abundance of microbial taxa, linear discriminant analysis effect size (LEfSe) [51] was employed for both analysis groupings. LefSe uses the Kruskal–Wallis test to identify taxa that are differentially abundant across groups and then performs a linear discriminant analysis (LDA) to estimate the effect size of each taxon’s contribution to the group differences. Random Forest analysis was also performed to identify features (bacterial genera) with the ability to best distinguish samples between PC and HC or IPMN and PDAC. A 500-tree iteration was used and a confusion matrix approach with the calculation of out-of-bag (OOB) error rates was implemented to evaluate the model. A confusion matrix and OOB error is a method used to measure the prediction error of Random Forests utilizing bootstrap aggregating (bagging). It is an estimate of the performance of a Random Forest classifier or regressor on unseen data. The OOB error is computed using the samples that were not included in the training of the individual trees, providing a measure of the model’s performance on a validation dataset by comparing true positive and negative results to the predicted ones. Taxonomic visualization, alpha- and beta-diversity analyses, LefSe, and machine learning calculations were carried out using Microbiome Analyst [52], on which data were rarefied to the smallest library size and scaled using total sum scaling after removing taxa with a low prevalence (in less than 10% of the samples).

## 3. Results

### 3.1. Taxonomic Differences

All three sample groups (controls, IPMN, and PDAC) exhibit distinct microbiome patterns based on their microbial taxa on different taxonomic levels. These disparities, overall, are more pronounced between HC and PC samples, while the PDAC and IPMN groups show similar patterns. On the phylum level, all HC samples are characterized by a dominance of *Bacteroidota* (60% relative abundance), followed by *Firmicutes* (33%) and *Proteobacteria* (5%), while the PC samples show a slight increase in *Firmicutes* (38%) and *Proteobacteria* (11%) and a significant decrease in *Bacteroidota* (46%), as shown in Figure 1a–c. The same three phyla are most abundant in the PDAC vs. IPMN comparison, without, however, exhibiting sizeable differences between the groups (IPMN: *Bacteroidota* 45%, *Firmicutes* 39%, *Proteobacteria* 12%, PDAC: *Bacteroidota* 47%, *Firmicutes* 38%, *Proteobacteria 11*%*)*, and they are presented in Figure 1d–f. On the family taxonomic level, the different microbial abundance patterns can be seen with the help of a heatmap representation (Figure 2), in which it is apparent that families like *Sutterrellaceae* and *Fusobacteriaceae* are almost non-detectable in HC samples but prominent in PC samples. The opposite pattern can be observed in microbial families like *Erysipelotrichaea*, *Akkermansiaceae*, and others, which are mainly detectable in the HC samples.

### 3.2. Microbial Diversity

Alpha-diversity metrics establish clear dysbiosis patterns between the HC and PC samples, presenting a clear loss of biodiversity both in raw taxa abundance and distribution. Chao1 (Figure 3a) and Shannon (Figure 3b) indices present these differences, while the FDR-corrected statistical significance between HC and PC samples is calculated to be adjusted-*p* = 7.4 × 10^−14^ and adjusted-*p* = 10 × 10^−15^, respectively. However, when comparing IPMN and PDAC samples, both Chao1 and Shannon indices fail to highlight statistically significant differences (adjusted-*p* = 0.71477 and adjusted-*p* = 0.913, respectively) (Figure 3c,d). 

Concerning beta-diversity, which provides a qualitative insight into the microbial composition of our samples, it is evident that while the PC samples share some characteristics with the HC samples, they exhibit greater dissimilarity among themselves, distinctly separating from the HC samples (ANOSIM R: 0.35, *p* < 0.001), as depicted in Figure 4a. However, there are no significant differences in microbial composition between the IPMN and PDAC samples (ANOSIM R: −0.008, *p* < 0.5, which can effectively be interpreted as *p* > 0.05) (Figure 4b).

### 3.3. The Microbiome as a Biomarker

Based on the LEfSe and Random Forest analyses performed, there is clear evidence that the microbiome can be effectively used to distinguish between samples that derive from pancreatic cancer patients and controls but falls short of differentiating between IPMN and PDAC. The LEfSe analysis highlights several microbial genera associated with PC, with the most pronounced being *Escherichia_Shigella* (two very genetically similar genera which SILVA cannot distinguish effectively, so it presents as one), *Fusobacterium*, *Sutterella*, *Klebsiella, Eubacterium_ventriosum*_group, *CAG_352*, *Bifidobacterium*, *Odoribacter*, *Eubacterium_ruminantium*_group, *Ezakiella*, and *Colidextribacter,* while *Bacteroides*, *Faecalibacterium*, *Agathobacter*, *Akkermansia*, *Subdoligranulum*, *Alistipes*, *Fusicatenibacter*, *Lachnospiraceae_UCG_004*, and *Lachnospira* are more abundant in HC samples (Figure 5a), with all achieving an effect size > 3 and FDR-adjusted *p* < 0.01. However, in the case of IPMN and PDAC, only *Lachnospira* abundance appears to be associated with IPMN, and *Ruminococcus_torques_group*, *Collinsella*, and *Family_XIII_AD3011*_group are more abundant in PDAC, without, however, achieving statistical significance (FDR-adjusted *p* > 0.1) (Figure 5b). The outcomes from LEfSe were further reinforced by the Random Forest analysis, which highlighted elevated abundance of *Butyrivibrio*, *Agathobacter*, *Hafnia_Obesumbacterium*, *Prevotellaceae_NK3B31_group*, *Methylobacterium_Methylorubrum*, *Barnesiella*, and *Ruminococcus_gnavus_group* as indicative markers for HC samples, while proposing *CAG_352* and *Lactobacillus* as potential biomarkers for PC. This analysis demonstrated high accuracy by correctly predicting all HC samples, with only one mislabeling incident for a PC sample, resulting in an Out-of-Bag (OOB) error of 0.0175 (Figure 6). However, the Random Forest model faced challenges in distinguishing and accurately predicting between IPMN and PDAC samples based on their microbial composition. It consistently characterized all samples as PDAC, leading to an OOB error of 1.0.

## 4. Discussion

As an insidious malignancy, PC remains a difficult challenge that requires extensive efforts for early detection to diminish its impact. Thus, there is a need to identify and develop novel clinical approaches and personalized treatments for effective PC management. Recently, the use of microbiome analysis has been accepted as a prognostic and diagnostic marker that holds numerous potential implications and advantages in clinical practice. As it is acknowledged that ethnicity-related variations in the gut microbiota likely signify differences in racial and environmental factors [53], our research aimed to explore the composition of the microbiome in Greek individuals with PC, a generally homogeneous population that also shares dietary and cultural habits with other Mediterranean people. In the present study, we found that patients with PC had higher abundances of *Firmicutes* and *Proteobacteria*, while HC had higher proportions of *Bacteroidota.* We did not, however, observe significant differences between the PDAC and IPMN groups (Figure 1). Our results are in agreement with previous studies on fecal and oral microbiota from PDAC and IPMN patients [18,54]. Interestingly, Mendez et al. [55], using a genetically engineered PDAC murine model, suggested that the increased abundance of *Proteobacteria* and *Firmicutes* in early PDAC is linked to an upregulation of the polyamine and nucleotide biosynthetic pathways, as well as with elevated serum polyamine concentration, findings that also have been verified in PDAC patients, suggesting a role of these bacteria in pancreatic carcinogenesis.

Another interesting finding of our study is the detection of *Akkermansiaceae* in PC samples (Figure 5). *Akkermansiaceae* has been associated with different cancers like lung cancer, renal cancer, bladder cancer, and prostate cancer [56,57] and has also been linked to immunotherapy response [58]. In addition, Kartal et al. [59] have detected an enrichment of *Akkermansia muciniphila* in PC samples. Regarding *Erysipelotrichaea,* which is also prominent in HC cases, it is known that is related with inflammation-related disorders of the gastrointestinal tract, such as colorectal cancer and hepatocellular carcinoma, and also associated with host lipid metabolism [60,61]. Recently, Half et al. [62] suggested that *Erysipelotrichaea* are correlated with enzyme GGT serum levels in PC patients. Even if several studies are contradictory regarding the gut microbial diversity in PC cases, because of the microbiota’s multifactorial perturbations, the results of our alpha- and beta-diversity analysis clearly indicate that the composition of the gut microbial population of PC patients is distinct from that of HC [19,62,63,64] (Figure 3 and Figure 4). Regarding the comparison between IPMN and PDAC cases, the diversity analysis did not show significant accuracy to distinguish PDAC patients from IPMN cases (Figure 3 and Figure 4). Olson et al. also reported that the oral microbiome in PDAC cases did not differ in diversity analysis from IPMN cases [18]. However, the current literature lacks multiple studies investigating the microbiome of these phenotypes, while public databases lack suitable microbiome samples. Furthermore, these limitations extend to the inability of current approaches to distinguish between PDAC and IPMN samples, as underscored by our Random Forest results (Figure 6), due to the high similarities their microbial composition presents.

Our results reveal a PC-associated microbial signature that can potentially be suggested as an effective biomarker (Figure 5). Among them, consistent with previous findings [65,66], bacteria like *Escherichia_Shigella*, *Klebsiella*, and *Fusobacterium* were enriched in PC-associated gut microbiota among other pro-inflammatory and cancer-promoting genera. Furthermore, our data confirm previous studies that *Lactobacillus* and *Bifidobacterium* are present in PDAC tumors [59]. *Lactobacillus* and *Bifidobacterium* spp, known producers of indole and/or indole lactic acid [67,68], have been linked to immunity modulation, oncogenesis in animal models, and poor outcomes in human PDAC. In support of this, Hezaveh et al. [69] suggested that indole-producing bacteria promote an immunosuppressive tumor microenvironment and correlate with poor response to resection and overall survival in PDAC. Additionally, *Lachnospira* appears enriched in IPMN cases compared to PDAC cases in our study. Members of the *Lachnospiraceae* family have been involved in carcinogenesis and it has been reported that they might influence colorectal cancer progression [70]. Also, in cases of acute pancreatitis, the relative abundance of *Lachnospira pectinoschiza* decreased in on-treatment samples compared with those before the treatment [71]. Finally, Vogtmann et al. have also reported an increase in the Lachnospiraceae in oral PDAC cases [72]. 

The high predictive power of the Random Forest analysis (Figure 6) between HC and PC samples provides promising outcomes on the application of the gut microbiome for early diagnosis. High abundances of *Butyrivibrio* and *Agathobacter*, two types of intestinal bacteria that produce butyrate, in HC samples versus PC samples further confirm the previous findings of the postbiotics’ ability to affect pancreatic cancer. Butyrate has been found to improve intestinal integrity and microbiota composition in pancreatic cancer mouse models [73,74]. In addition, low levels of butyrate have also been correlated with high levels of acetate in patients with adenomatous polyp formation and colon cancer [75]. While *Butyrivibrio fibrisolvens* specifically has been found to suppress cancer-associated fibroblasts in pancreatic cancer [76], we could not assess the presence of this species in our samples due to the limitations of the 16S approach, but our results might signify its existence.

In general, the 16S approach for discerning microbial species is constrained by technical limitations, despite being a fast and easily accessible testing method for biomarker discovery. For example, in this study, a moderate amount of reads per sample, with a typical read length for the technology, hinders high resolution and accurate species-level identification. In general, high-throughput sequencing of the amplicons of hypervariable 16S rRNA gene regions has been a mainstay for bacterial analysis, but it has limitations in discerning species and strain-level diversity. While it can be used to identify and compare bacterial diversity from complex microbiomes, the method’s ability to provide accurate and complete sequences is essential for its utility in many applications [77,78]. However, third-generation sequencing platforms which can provide full-length 16S amplification in microbiome studies have seen rise in recent years, providing several advantages over short-read sequencing, including higher resolution in terms of diversity and taxonomic classification and the ability to detect additional taxa that may be missed by short-read sequencing [79]. Full-length 16S sequencing has been shown to provide species-level resolution in human gut microbiota studies [80]. However, full-length 16S sequencing also has limitations, including a higher cost and longer analysis time, and still may not be able to discriminate some closely related species [81]. This also hinders our ability to detect the metabolic contributions of the microbiome to host physiology, without inferences, and obstructs any assumptions regarding the effects of the microbiome in metabolomics. 

Regardless of the issues and complexities of microbiome research, we now know that it has the potential to have real clinical implications, even from a pharmacological standpoint. The gut microbiota affects the occurrence and development of cancer, along with the efficacy and toxicity of chemotherapy, radiotherapy, and immunotherapy [82]. Modulating the gut microbiota has been proposed as a potential strategy for cancer prevention and treatment [83,84,85]. Bacteria can also be used to bypass problems associated with the poor selectivity and limited tumor penetrability of conventional cancer therapies or can be engineered to directly express anticancer agents or transfer eukaryotic expression vectors to cancer cells [82].

Overall, this study, despite its limitations in sample-pool size and the lack of addressing confounding factors, provides evidence of the gut microbiome’s ability to serve as a biomarker of PC detection and outlines the steps needed to further distinguish between its different phenotypes. The microbiota, as we have already discussed, are very sensitive to a plethora of factors, and their variability from person to person, even between biological sexes [86], constitutes the complexity of microbiome studies. Unfortunately, the small sample size prohibits us from further segmenting the dataset to account for sex, although we did take measures to collect a balanced number of male and female participants for PDAC and controls. The same was not possible for IPMN due to the paucity of samples. 

A study of this magnitude can only offer insights into microbial variances and is not a conclusive method for clinical diagnosis. In line with previous studies, investigations like ours can only provide statistical differences between sample groups. The ability of the microbiome to serve as a diagnostic marker relies on those statistical differences exhibited here, while prognostically it can serve as a biomarker if dysbiosis precedes the diagnosis. To be certain of that, we would need a much larger cohort with random samples of people who have not been diagnosed yet; however, the presence of a “cancer”-specific microbiome might serve as a prognostic indicator. Nevertheless, we maintain that in the realm of microbiome research, a consensus derived from smaller, more manageable studies can progressively enhance our comprehension of the microbial underpinnings of these conditions. Our results further enrich current knowledge of this formidable cancer type, while showcasing the practical utility of microbiome research.

## Figures and Tables

**Figure 1 biomedicines-12-01040-f001:**
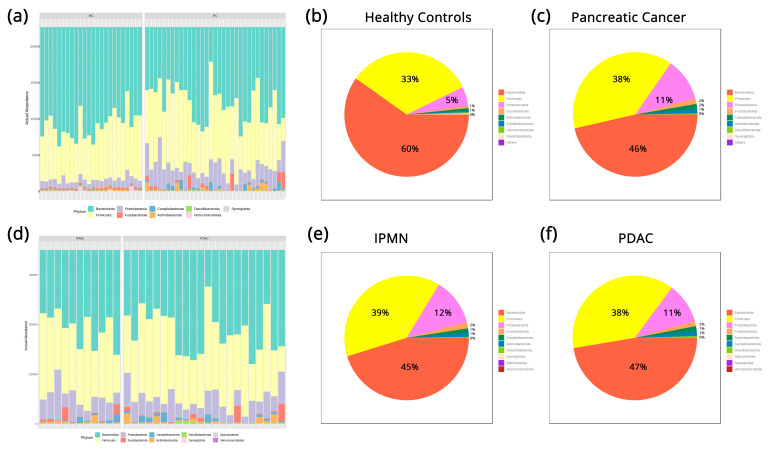
(**a**) Taxonomic profiles of all samples on the Phylum level. (**b**) Pie charts of Phylum relative abundance in healthy control samples. (**c**) Pie charts of Phylum relative abundance in pancreatic cancer samples. (**d**) Taxonomic profiles of all pancreatic cancer samples separated by subtype on the Phylum level. (**e**) Pie charts of Phylum relative abundance in IPMN samples. (**f**) Pie charts of Phylum relative abundance in PDAC samples.

**Figure 2 biomedicines-12-01040-f002:**
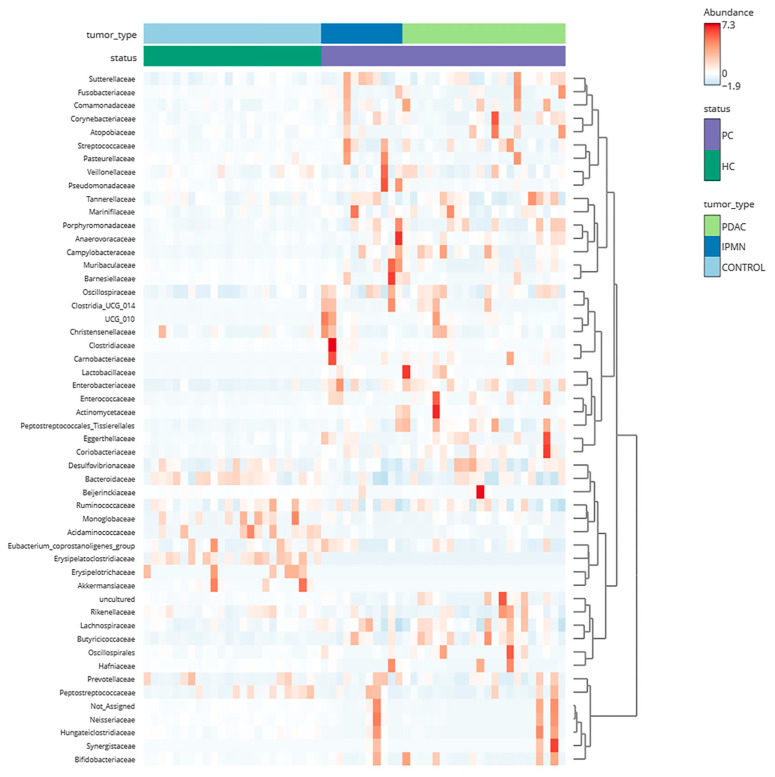
Heatmap of microbial family abundance in all samples categorized for cancer presence and phenotypes.

**Figure 3 biomedicines-12-01040-f003:**
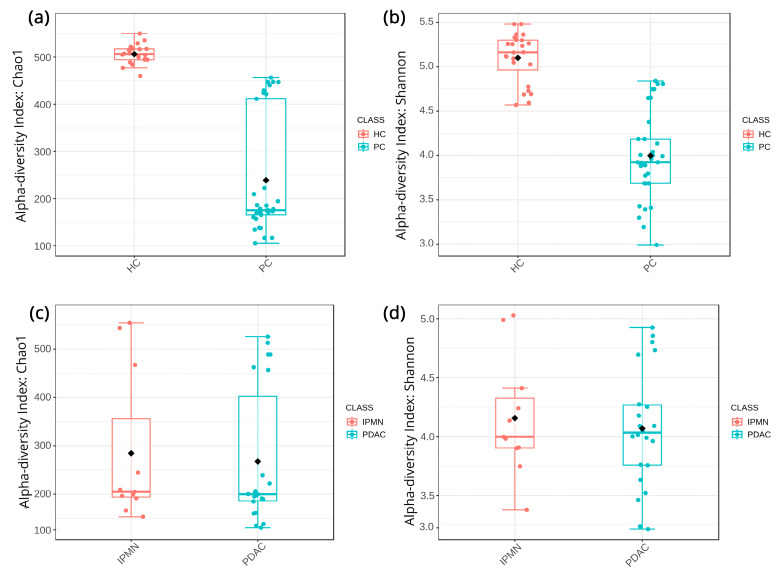
(**a**) Chao1 index alpha-diversity between healthy controls and pancreatic cancer samples. (**b**) Shannon index alpha-diversity between healthy controls and pancreatic cancer samples. (**c**) Chao1 index alpha-diversity between IPMN and PDAC samples. (**d**) Shannon index alpha-diversity between IPMN and PDAC samples. For all boxplots parallel line represents the median while the black dot is the mean.

**Figure 4 biomedicines-12-01040-f004:**
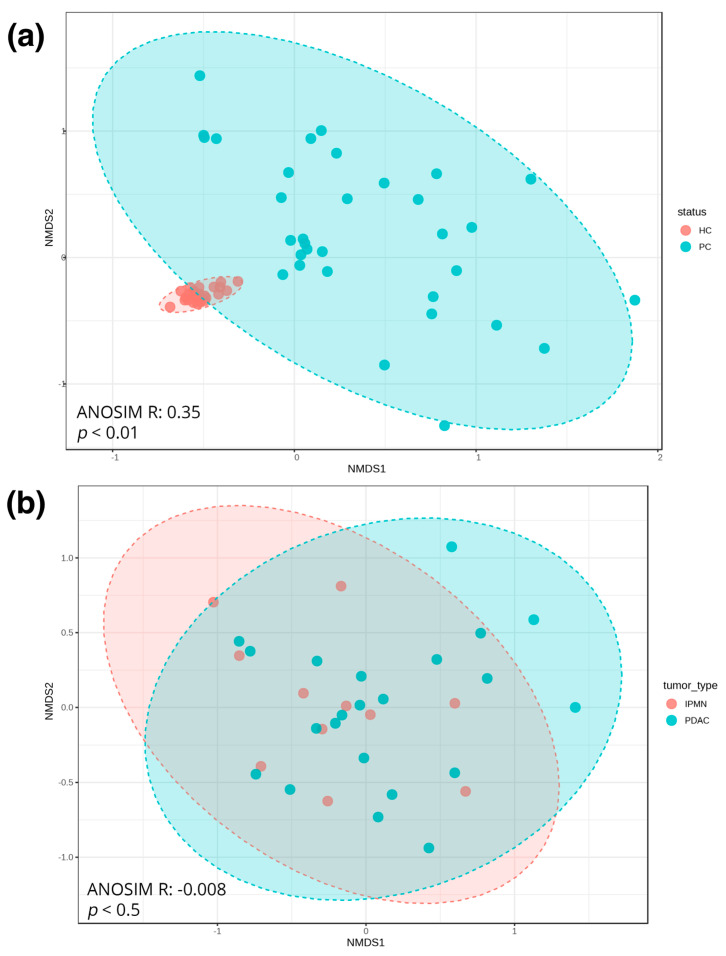
(**a**) NMDS graphs of beta-diversity between healthy controls and pancreatic cancer samples. (**b**) NMDS graphs of beta-diversity between IPMN and PDAC samples.

**Figure 5 biomedicines-12-01040-f005:**
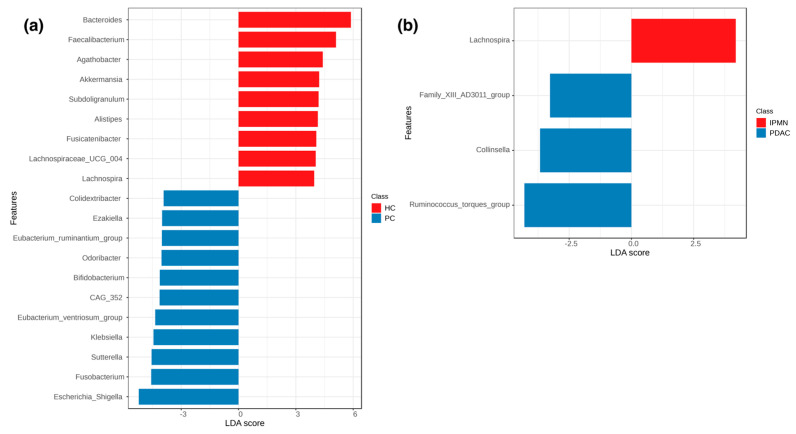
(**a**) LeFSe diagram of linear discriminant analysis scores for microbial genera between healthy control and pancreatic cancer samples. (**b**) LeFSe diagram of linear discriminant analysis scores for microbial genera between IPMN and PDAC samples.

**Figure 6 biomedicines-12-01040-f006:**
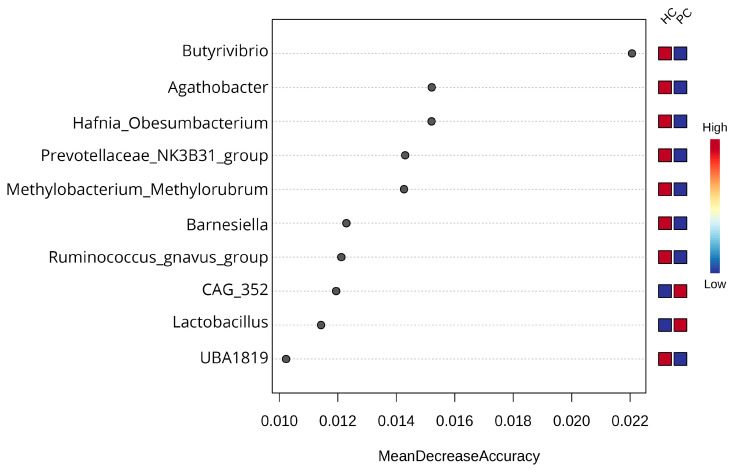
Important features (microbial genera) selected by the Random Forest model to discern between healthy control and pancreatic cancer samples.

**Table 1 biomedicines-12-01040-t001:** Clinicopathological data of the patients and controls.

Characteristics	PDAC (*n* = 22)	IPMN (*n* = 11)	Controls (*n* = 24)
Mean age ± SD, years	66.75 ± 13.40	67.36 ± 7.67	57.21 ± 17.20
Sex			
Male	10	9	13
Female	12	2	11
Smoking			
Yes	13	6	Not Available
No	9	5	Not Available
Tumor stage			
I	5		
II	5		
III	7		
IV	5		
Tumor location			
Head	6	2	
Tail	3	-	
Body	13	9	

SD: standard deviation.

## Data Availability

Data are contained within the article and Appendix A.

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
