# Peer review of "Dysbiosis Signature of Fecal Microbiota in Patients with Pancreatic Adenocarcinoma and Pancreatic Intraductal Papillary Mucinous Neoplasms"

_biomedicines, 2024, doi:10.3390/biomedicines12051040_

Round 1

Reviewer 1 Report

Comments and Suggestions for Authors

The authors presented an analysis of fecal microbiome in pancreatic cancer vs healthy patients. The authors successfully identified specific group microbiomes that are dysregulated in disease state patients. At last, authors used machine learning methods to explore the possibilities to use microbiome profile as diagnostic biomarker for pancreatic ductal cancers. In the discussion, authors also indicated potential caveats of the study. I found the whole manuscript well prepared and scientifically sounding. Only minor suggestion is the formatting of the figures. Texts and axis labels are very small and not aligned. For example, Figure 3 axis labels are misaligned, and some numbers were covered by bigger figure legends. 

Author Response

Comment:

The authors presented an analysis of fecal microbiome in pancreatic cancer vs healthy patients. The authors successfully identified specific group microbiomes that are dysregulated in disease state patients. At last, authors used machine learning methods to explore the possibilities to use microbiome profile as diagnostic biomarker for pancreatic ductal cancers. In the discussion, authors also indicated potential caveats of the study. I found the whole manuscript well prepared and scientifically sounding. Only minor suggestion is the formatting of the figures. Texts and axis labels are very small and not aligned. For example, Figure 3 axis labels are misaligned, and some numbers were covered by bigger figure legends.

Response:

We would like to thank the reviewer for their time and efforts. We appreciate their interest and approval of our manuscript. Figure 3 has been recreated according to their comments.

Reviewer 2 Report

Comments and Suggestions for Authors

The manuscript titled "Dysbiosis Signature of Fecal Microbiota in Patients with Pancreatic Adenocarcinoma and Pancreatic Intraductal Papillary Mucinous Neoplasms" presents an investigation into the fecal microbiota dysbiosis signatures among pancreatic cancer (PC) patients, focusing on differentiating between Pancreatic Ductal Adenocarcinoma (PDAC) and Intraductal Papillary Mucinous Neoplasm (IPMN) cases.

The study’s relevance is well-founded, given the significant clinical challenge pancreatic cancer represents and the potential for microbiome signatures to aid early detection.

1.     The study addresses an important and timely question in cancer research—using the microbiome as a diagnostic tool for pancreatic cancer

2.     The use of both LEfSe and Random Forest analyses for biomarker discovery is methodologically sound, providing a robust approach to data analysis.

3.     The study includes a relatively small sample size (n=33), which might limit the generalizability of the findings. Additionally, the study population appears to be ethnically homogeneous (Greek), which might not translate to other populations due to known microbiome variability across different ethnicities and diets.

4.     Despite the use of sophisticated analytical tools, the study fails to distinguish between microbiota signatures of PDAC and IPMN effectively. This finding might limit the clinical utility of the microbiome as a diagnostic tool unless further refined. However, it’s important also to publish “negative results”

Minor Concerns:

  1. Some details about the sequencing depth and data processing could be more explicitly detailed to ensure reproducibility. The discussion on sequencing platforms' limitations and the potential need for newer technologies to enhance microbiome studies is appreciated but needs to be connected more explicitly to the study’s limitations in this regard.
  2. While the statistical tools used are appropriate, the study could benefit from a more detailed explanation of the statistical power and effect size, particularly in light of the small sample sizes for PDAC and IPMN.
  3. Please provide more detailed descriptions of the methodologies for microbiome analysis, including any data normalization processes and detailed statistical analysis methodologies
  4. A more detailed characterization of the study population, considering potential confounders that might affect the microbiome, such as dietary habits, medication use, and prior ERCP, choledocolithiasis (see  PMID: 38637624) would strengthen the conclusions drawn.
  5. Some figures and tables are discussed extensively in the text, but clearer integration into the discussion could help in better visualization and understanding of the data presented. Specifically, the heatmap representations and taxonomic profiles could be more directly linked to the discussion on microbiome variability.
  6. In the introduction please add these citations (PMID: 36766711, PMID: 38622651, PMID: 38540258, PMID: 35203339)
Comments on the Quality of English Language

The manuscript is generally well-written, even if some areas can be improved to enhance readability and clarity. Only as an example: the consistent use of either "microbiota" or "microbiome" throughout the document could help maintain consistency, as these terms are sometimes used interchangeably but can imply different aspects of the microbial community

Author Response

We would like to thank the reviewer for their time and efforts. Please find below detailed answers to your comments along with the amendments to our manuscript.

Comment:

The manuscript titled "Dysbiosis Signature of Fecal Microbiota in Patients with Pancreatic Adenocarcinoma and Pancreatic Intraductal Papillary Mucinous Neoplasms" presents an investigation into the fecal microbiota dysbiosis signatures among pancreatic cancer (PC) patients, focusing on differentiating between Pancreatic Ductal Adenocarcinoma (PDAC) and Intraductal Papillary Mucinous Neoplasm (IPMN) cases.

The study’s relevance is well-founded, given the significant clinical challenge pancreatic cancer represents and the potential for microbiome signatures to aid early detection.

Response:

Thank you. We appreciate the positive feedback on our hypothesis and investigation

Comment:

The study addresses an important and timely question in cancer research—using the microbiome as a diagnostic tool for pancreatic cancer

Response:

We share the reviewer's views on the importance of the microbiome as a diagnostic tool.

Comment:

The use of both LEfSe and Random Forest analyses for biomarker discovery is methodologically sound, providing a robust approach to data analysis.

Response:

Thank you, that was our goal. Microbiome analysis is a complex and multifaceted ordeal that in silico needs multiple methods to add value to any results.

Comment:

The study includes a relatively small sample size (n=33), which might limit the generalizability of the findings. Additionally, the study population appears to be ethnically homogeneous (Greek), which might not translate to other populations due to known microbiome variability across different ethnicities and diets.

Response:

This is true. The sample size can be considered rather small for any generalizations and that’s why we have highlighted this fact in the discussion. In our opinion, a homogeneous population is the optimal way to investigate the microbiome due to its large influence by a significant amount of environmental, dietary, ancestry and geographical factors.  However, studies have shown that even though the exact microbial composition might vary from country to country microbial function presents many similarities.

Comment:

Despite the use of sophisticated analytical tools, the study fails to distinguish between microbiota signatures of PDAC and IPMN effectively. This finding might limit the clinical utility of the microbiome as a diagnostic tool unless further refined. However, it’s important also to publish “negative results”

Response:

Again, we wholeheartedly agree with the reviewer. We cannot be sure that there are no discernible differences in the microbiome of PDAC and IPMN due to their nature or because of our small sample size.  However, previous studies have also shown little to no differences, as discussed, perhaps due to the fact of the paucity of IPMN samples available for investigation in general.

Comment:

Some details about the sequencing depth and data processing could be more explicitly detailed to ensure reproducibility. The discussion on sequencing platforms' limitations and the potential need for newer technologies to enhance microbiome studies is appreciated but needs to be connected more explicitly to the study’s limitations in this regard.

Response:

Per the reviewer’s suggestion we have added more information regarding the sequence output (lines 133-136) and have commented on the limitation imposed on this study by the choice of amplicon sequencing (lines 351-353).

Comment:

While the statistical tools used are appropriate, the study could benefit from a more detailed explanation of the statistical power and effect size, particularly in light of the small sample sizes for PDAC and IPMN.

Response:

Unfortunately, we haven’t performed any statistical power analysis while designing this study to provide more details. The small sample size was not driven by design but rather by the availability of clinical samples. We relied solely on the statistical power of individual tests (for example p-adjusted values as reported for a-diversity and linear effect size for LEfSe).

Comment:

Please provide more detailed descriptions of the methodologies for microbiome analysis, including any data normalization processes and detailed statistical analysis methodologies

Response:

The methodology section has been amended to include any pre-processing steps including normalization along with the detailed statistical testing information already provided (Lines:  148-154)

Comment:

A more detailed characterization of the study population, considering potential confounders that might affect the microbiome, such as dietary habits, medication use, and prior ERCP, choledocolithiasis (see  PMID: 38637624) would strengthen the conclusions drawn.

Response:

The exclusion criteria listed in lines 108-112  provide all available medication information we had access to. In addition, line 102 states that some patients had undergone MRCP but unfortunately we have no information on  choledocolithiasis. All patients as stated were treatment-naïve for cancer. Dietary habits include standard Mediterranean diet with the addition of more contemporary foods (fast-food etc) which are common for a Greek population. The manuscript has been amended to show that no  dietary restrictions were imposed prior to this study.

Comment:

Some figures and tables are discussed extensively in the text, but clearer integration into the discussion could help in better visualization and understanding of the data presented. Specifically, the heatmap representations and taxonomic profiles could be more directly linked to the discussion on microbiome variability.

Response:

As per the reviewer’s recommendation, we have supplemented the discussion with references to specific figures to assist with the readability of the manuscript.

Comment:

In the introduction please add these citations (PMID: 36766711, PMID: 38622651, PMID: 38540258, PMID: 35203339)

Response:

As per the reviewer’s recommendation, the references have been added appropriately in the introduction.

Comment:

The manuscript is generally well-written, even if some areas can be improved to enhance readability and clarity. Only as an example: the consistent use of either "microbiota" or "microbiome" throughout the document could help maintain consistency, as these terms are sometimes used interchangeably but can imply different aspects of the microbial community

Response:

We have gone through the whole manuscript to amend misuse of the words microbiota and microbiome to reflect where we refer to the microorganisms (microbiota) or their genetic composition (microbiome)

Reviewer 3 Report

Comments and Suggestions for Authors

The manuscript "Dysbiosis signature of fecal microbiota in patients with pancreatic adenocarcinoma and pancreatic intraductal papillary mucinous neoplasms" is well written and the conclusion is supported by the data provided. The following concerns should be addressed.

1. Line 109: aby should be any.

2.  The sample population consists of male and female gender and sex difference may have an effect on microbiota (https://doi.org/10.1016/j.yfrne.2021.100912). Please comment in the discussion and if possible, please analyze the data considering the sex as a dependent factor.

3.Please include text on how the microbiota identified in the article can affect the metabolomics promoting PDAC formation from PANINs.

4. What is the prognostic value of the microbiota identified in the study (prognostic and diagnostic biomarkers).

5. Please include recent studies suggesting microbiota as biomarker in PDAC.

6. Please include the translational aspect- how the identified biomarkers may serve as therapeutic targets.

Author Response

We would like to thank the reviewer for their thorough efforts in elevating our manuscript and improving the reader’s experience. Please find below the point-by-point answers to your comments.  

Comment:

The manuscript "Dysbiosis signature of fecal microbiota in patients with pancreatic adenocarcinoma and pancreatic intraductal papillary mucinous neoplasms" is well written and the conclusion is supported by the data provided. The following concerns should be addressed.

Response:

Thank you!

Comment:

The sample population consists of male and female gender and sex difference may have an effect on microbiota (https://doi.org/10.1016/j.yfrne.2021.100912). Please comment in the discussion and if possible, please analyze the data considering the sex as a dependent factor.

Response:

Unfortunately, the small sample size prohibits us from further segmenting the dataset to account for gender. However, as you can see from Table 1 we had taken measures of collecting a balanced number of male and female participants for PDAC and controls. The same was not possible for IPMN due to the paucity of samples. In addition, we have amended the discussion section to address this subject along with the provided reference.

Comment:

Please include text on how the microbiota identified in the article can affect the metabolomics promoting PDAC formation from PANINs

Response:

As discussed in the manuscript there are certain limitations in using 16S amplicon sequence for microbiome research. One of these limitations is the fact that we cannot detect the metabolic contributions of the microbiome to host physiology. With that in mind making any assumptions regarding the effects of the microbiome in metabolomics is, unfortunately, impossible.

Comment:

What is the prognostic value of the microbiota identified in the study (prognostic and diagnostic biomarkers).

Response:

In line with other studies, investigations like ours can only provide differences between sample groups. The ability of the microbiome to serve as a diagnostic marker relies on those statistical differences exhibited here while prognostically might be able to serve as a biomarker if dysbiosis precedes the diagnosis. To be certain of that we would need a much larger cohort with random samples of people who haven’t been diagnosed yet, however, the presence of a “cancer” specific microbiome might serve as a prognostic indicator.

Comment:

Please include recent studies suggesting microbiota as biomarker in PDAC.

Response:

The manuscript has been enriched with more references to microbiome research efforts to serve as a biomarker for PDAC

Comment:

Please include the translational aspect- how the identified biomarkers may serve as therapeutic targets.

Response:

The discussion section has been expanded to include the targeting and use of microbiota in cancer therapeutic practices.

Round 2

Reviewer 2 Report

Comments and Suggestions for Authors

The authors have diligently addressed all the questions and concerns raised in the previous reviews.

Author Response

Once more, we appreciate all the reviewer's efforts in improving our manuscript. 

Reviewer 3 Report

Comments and Suggestions for Authors

Please include these responses in the "Limitations of Study" section

Unfortunately, the small sample size prohibits us from further segmenting the dataset to account for gender. However, as you can see from Table 1 we had taken measures of collecting a balanced number of male and female participants for PDAC and controls. The same was not possible for IPMN due to the paucity of samples. In addition, we have amended the discussion section to address this subject along with the provided reference.

As discussed in the manuscript there are certain limitations in using 16S amplicon sequence for microbiome research. One of these limitations is the fact that we cannot detect the metabolic contributions of the microbiome to host physiology. With that in mind making any assumptions regarding the effects of the microbiome in metabolomics is, unfortunately, impossible.

In line with other studies, investigations like ours can only provide differences between sample groups. The ability of the microbiome to serve as a diagnostic marker relies on those statistical differences exhibited here while prognostically might be able to serve as a biomarker if dysbiosis precedes the diagnosis. To be certain of that we would need a much larger cohort with random samples of people who haven’t been diagnosed yet, however, the presence of a “cancer” specific microbiome might serve as a prognostic indicator.

Author Response

Thank you, again, for your invaluable input. We have amended the manuscript as suggested and uploaded the new version.